# Treating (Recurrent) Vulvovaginal Candidiasis with Medical-Grade Honey—Concepts and Practical Considerations

**DOI:** 10.3390/jof7080664

**Published:** 2021-08-16

**Authors:** Senna J. J. M. van Riel, Celine M. J. G. Lardenoije, Guy J. Oudhuis, Niels A. J. Cremers

**Affiliations:** 1Department of Gynecology and Obstetrics, Maastricht University Medical Centre, 6202 AZ Maastricht, The Netherlands; sjjmvanriel@gmail.com (S.J.J.M.v.R.); celine.lardenoije@mumc.nl (C.M.J.G.L.); 2Department of Medical Microbiology, Maastricht University Medical Centre, NUTRIM School of Nutrition and Translational Research in Metabolism, Faculty of Health, Medicine and Life Sciences, Maastricht University, 6202 AZ Maastricht, The Netherlands; g.oudhuis@mumc.nl; 3Triticum Exploitatie B.V., Sleperweg 44, 6222 NK Maastricht, The Netherlands

**Keywords:** recurrent vulvovaginal candidiasis, medical-grade honey, fluconazole, alternative treatment, micro-environment modulation

## Abstract

Recurrent vulvovaginal candidiasis (RVVC) is a relapsing vaginal fungal infection caused by *Candida* species. The prevalence varies among age populations and can be as high as 9%. Treatment options are limited, and in 57% of the cases, relapses occur within six months after fluconazole maintenance therapy, which is the current standard of care. The pathogenesis of RVVC is multifactorial, and recent studies have demonstrated that the vaginal microenvironment and activity of the immune system have a strong influence on the disease. Medical-grade honey (MGH) has protective, antimicrobial, and immunomodulatory activity and forms a putative alternative treatment. Clinical trials have demonstrated that honey can benefit the treatment of bacterial and *Candida*-mediated vaginal infections. We postulate that MGH will actively fight ongoing infections; eradicate biofilms; and modulate the vaginal microenvironment by its anti-inflammatory, antioxidative, and immunomodulatory properties, and subsequently may decrease the number of relapses when compared to fluconazole. The MGH formulation L-Mesitran Soft has stronger antimicrobial activity against various *Candida* species than its raw honey. In advance of a planned randomized controlled clinical trial, we present the setup of a study comparing L-Mesitran Soft with fluconazole and its practical considerations.

## 1. Introduction

Vulvovaginal candidiasis (VVC) is a vaginal fungal infection confirmed to be caused by *Candida* species, in most cases *Candida albicans* [1]. A total of 75% of women develop vulvovaginal candidiasis at least once in their life [2]. The symptoms related to vulvovaginal candidiasis are pruritus, soreness, irritation, dyspareunia, vaginal discharge, and discomfort. Clinical signs are exemplified by vulva erythema, edema, excoriation, and fissure formation, together with introital and vaginal erythema [1,3]. A non-malodorous clumpy white discharge is suggestive of VVC but is extremely nonspecific [1]. Women also report loss of confidence and self-esteem, inability to carry on with their normal physical activities, and difficulties with their sexual life and intimate relationships [2]. It also has a profound effect on the quality of life of affected women with additional systemic symptoms including depression and anxiety [1]. The definition of recurrent vulvovaginal candidiasis (RVVC) is at least three symptomatic episodes in the last 12 months [1]. RVVC affects about 138 million women per year worldwide (range of 103–172 million), with a global annual prevalence of 3871 per 100,000 women [2]. The highest prevalence (9%) is seen in women of an age between 25 and 34 years old. It is estimated that the population of women with recurrent vulvovaginal candidiasis will increase to almost 158 million in 2030 [2].

RVVC is a multifactorial disease whose symptoms are governed by the interaction between *Candida* (species and virulence factors), the *Lactobacilli* population, the micro-environment (inflammatory status, oxidative stress, estrogen), and the host (immune status, behavioral factors, genetic factors). A disbalance in any of these factors may induce RVVC [4].

VVC is according to the Clinical Practice Guidelines treated with topical or oral antifungals, of which azoles (miconazole, clotrimazole, and fluconazole) are the most commonly prescribed [5]. Notably, there is an increase in resistance of *Candida* species towards antifungal agents, causing multidrug resistance to emerge, while the long-term efficacy of antifungal agents is limited [3,6,7]. Therefore, an urgent need for alternative or complementary therapies to effectively treat RVVC and prevent it from recurrence is needed. It is important to know more about the etiology of RVVC, the different treatment options, and their efficacy to understand how novel therapies could improve the clinical outcome and quality of life.

## 2. Diagnosis of RVVC

RVVC is often overdiagnosed or misdiagnosed when the diagnosis is only based on clinical symptoms, which are non-specific. Laboratory testing is necessary to confirm the diagnosis of VVC, because self-diagnosis based on symptoms has an accuracy rate of only 28% for *Candida albicans* in self-treating women, making over-the-counter (OTC) antifungals often ineffective [5,8,9]. The golden standard for the diagnosis of VVC is by culturing the cells. It is also possible to use microscopy to identify yeast cells and hyphae. Gram staining of vaginal discharge mixed with potassium hydroxide (KOH) is used to distinguish *Candida* yeast cells and hyphae, which is relevant to the stage of the pathogenesis. The pH of the vaginal discharge is also an important indicator and normally stays within a range of 4.0–4.5. A pH above 4.7 is indicative of other infections such as bacterial vaginosis, trichomoniasis, or mixed infections [1,10,11]. For further differentiation between *Candida* species, additional culturing is needed, e.g., with chromogenic agar or Sabourad’s dextrose agar. However, culturing is not the most selective procedure, and molecular methods such as sequencing of the internal transcribed spacer (ITS sequencing) or matrix-assisted laser desorption ionization time-of-flight mass spectrometry (MALDI-ToF MS) are needed to identify the specific species [12,13]. This is especially relevant in the case of RVVC, in which non-albicans species (NAC) are becoming more prevalent. Similarly, susceptibility testing may be of adjuvant need in RVVC, as these infections are more resistant to antifungal agents [11].

## 3. Pathogenesis of RVVC

In approximately 90% of VVC episodes, *Candida albicans* is the causative agent [14]. However, NAC species can also cause VVC, such as *Candida glabrata*, *Candida krusei*, *Candida parapsilosis*, *Candida tropicalis*, and *Candida dubliniensis*, with *Candida glabrata* predominating [15].

Asymptomatic colonization of *Candida* species may persist for years because yeasts can live in symbiosis with vaginal microbiota and are tolerated by the immune system [1,16]. Acute symptomatic VVC causes a breakdown in the symbiotic relationship and is caused by an overgrowth of *Candida* or alteration in the host protective defense mechanisms [1]. Often, there is an underlying cause of the imbalance (e.g., antibiotic therapy) that allows *Candida* species to overgrow [1,2,4]. There seems to be a local mucosal overreaction caused by an exaggerated inflammatory response responsible for vulvovaginal symptoms [1,4]. This response can be triggered by host pattern recognition receptors (PRP) interacting with fungal pathogen-associated molecular patterns (PAMPs) and other more complex mechanisms (i.e., secreted aspartyl proteases (Sap)-mediated NLRP3 activation and the cytolytic peptide toxin candidalysin) [17,18]. The activation of the innate immune system by a series of proinflammatory cytokines and chemokines leads to the recruitment of neutrophils to the vaginal mucosa, subsequently exacerbating symptomatic disease [17,18,19]. Moreover, *Candida* species also can exert tissue damage by direct invasion of its hyphal filaments or secretion of virulence effectors [19].

*Candida species* have a dual lifestyle as both a vaginal commensal and an opportunistic pathogen [20]. The switch between yeast and hyphal growth is critical for virulence, affecting numerous properties including phenotypic and biochemical properties [18,21]. Virulence traits are created by microorganisms and may subsequently lead to tissue damage, making them more pathogenic [22]. Virulence traits include the ability to make a morphological switch from yeast to hyphae, modulating the expression of adhesins to help the *Candida* adhere to epithelial cells, the expression of invasins, the formation of biofilms, the secretion of hydrolytic enzymes such as Sap and candidalysin, and the ability to escape from phagocytosis by neutrophils and macrophages [17,18,23,24,25].

The pathogenesis of VVC is a gradual process that goes through several stages, from adherence to the vaginal epithelium, recognition (caused by a burden threshold of hyphae), becoming invasive, the possibility of biofilm formation, and dispersion of planktonic cells that reinitiates the complete process (see Figure 1).

### 3.1. Adhesion

The most important phase of the initial contact between potential pathogen and host is the adherence of yeast cells to epithelial cells. Consequently, an interaction follows between epithelial receptors and *Candida* adhesins. These *Candida* adhesins vary depending on the morphological status of the *Candida*, i.e., yeast or hyphae [26]. After this initial contact, hyphae will grow out of the yeast cells, which is the first step in the pathogenesis [27]. The progress of the infection is determined by the hyphal-expressed adhesins that play a key role in the pathogenesis [26]. In most cases, yeast cells adhere directly to the epithelial cells; however, there may also be direct adhesion of the hyphal form of *Candida* to epithelial cells, as hyphae grow from one epithelial cell to an adjacent epithelial cell [26]. After adherence to the vaginal epithelium, the *Candida* cells may form hyphae and become recognized by the immune system [26].

### 3.2. Recognition 

The fact that *Candida* species have a dual lifestyle as both a vaginal commensal and opportunistic pathogen requires epithelial cells to have a mechanism to discriminate between the colonizing yeast and the invasive hyphal form [20]. *Candida* species can be recognized via two different pathways. The first phase is represented by an early transient response that occurs in a morphologically independent manner. The second phase starts when a burden threshold of hyphae is reached; this is a stronger response to hyphae, which in turn leads to activation of epithelial cells and the production of cytokines, chemokines, and other inflammatory mediators [26]. The hyphal burden plays a significant role in affecting epithelial activation because the presence of hyphae goes undetected below a certain threshold level. This is supported by in vivo experiments in a murine model of vaginitis, where the hyphal form was needed to cause damage to the epithelium, releasing pro-inflammatory cytokines and neutrophil recruitment [26]. Moreover, healthy women who were challenged intravaginally with *Candida* showed a differential susceptibility to develop symptomatic VVC when having no history of VVC (15%) compared to women with a history of infrequent VVC (55%) [28].

### 3.3. Invasion 

Invasion of *Candida* species can occur with the help of invasins after a switch from the yeast to the hyphal form [27,29]. Invasion into host cells can be achieved by endocytosis or active penetration [29]. This consequently leads to damage to the epithelial cells via necrosis and apoptosis, hence loss of epithelium. An in vivo study demonstrated a significantly lower cell damage caused by non-hyphal mutants compared to hyphal mutants [26,30].

### 3.4. Biofilms

Biofilm formation may be the main etiological factor contributing to antifungal resistance and is a likely contributor to treatment failure in RVVC [27,31,32,33]. It is suggested that the ability to form biofilms is a major virulence trait of *Candida* species in the pathogenesis of VVC [32,34]. After the invasion stage, an extracellular matrix (ECM) may be developed, which results in a ‘biofilm’ that encapsulates the *Candida* cells. This ECM consists of exopolymeric macromolecules, including polysaccharides, proteins, lipids, and nucleic acids, which are secreted by sessile cells within the biofilm. A mature biofilm is characterized by a structured mixture of yeast-form and hyphal cells surrounded by ECM, and it provides protection for the yeast against environmental challenges. In the final stage, the biofilm slowly disperses yeast-form cells into the surrounding, which may be able to colonize other surfaces [20,31].

Several studies confirm the role of biofilms in the pathogenesis of VVC. An in vitro study showed the ability of five types of *Candida* species isolated from patients with VVC, including *Candida albicans* and *Candida glabrata*, to form biofilms [35]. A study performed on clinical isolates from 300 women with at least two episodes of VVC demonstrated the ability to form heterogeneous biofilms [36]. In vivo and ex vivo murine vaginitis models confirmed the presumption of *Candida albicans* biofilm formation on vaginal epithelial cells, possibly due to the high fungal burden [33,36]. *Candida albicans* biofilms were also formed on reconstituted human vaginal epithelial cells [33,37]. Although the role of biofilms in clinical VVC remains disputed, many researchers consider them as important drivers of disease [32,33,36,38]. More research is needed to elucidate the role of *Candida* biofilms in the pathogenesis of (R)VVC [33].

## 4. Risk Factors of RVVC

VVC is considered a multifactorial disease, where the *Candida* strain and its virulence factors, an imbalanced vaginal microbiota composition, host-related predisposing factors, and idiopathic factors determine the disease onset and relapsing property [4].

### 4.1. Imbalanced Vaginal Microbiota Composition

Alteration in the mucosal ecosystem leading to fungal dysbiosis can lead to (R)VVC and its symptoms [4]. A healthy vaginal microflora consists of different microorganisms, mainly *Lactobacilli*, but also accommodating fungi such as *Candida albicans* and *Candida glabrata*, living in symbiosis. *Lactobacilli* species play an important role in maintaining a healthy vaginal microbiome [4]. Through their presence, *Lactobacilli* species decrease opportunism of potentially pathogenic microorganisms by microbial competition, which reduces the adherence of *Candida* species to the vaginal epithelium [24,39]. *Lactobacilli* exert several other beneficial properties, such as decreasing the vaginal pH, which in healthy women of reproductive age remains between 4 and 4.5. As the fermentation of glycogen within the vaginal epithelium produces lactic acid, the vaginal pH decreases and non-resident microbiota are suppressed, thereby protecting against opportunistic infections [27,39,40,41]. In addition, *Lactobacilli* produce bacteriocins and hydrogen peroxide (H_2_O_2_), which positively affect the commensal community by limiting pathogens through their antimicrobial function [39]. Moreover, *Lactobacilli* species induce the expression of genes that prevent the adherence of yeast to the epithelium and limit yeast-to-hyphal formation, keeping the *Candida* in its less invasive form and consequently inhibiting overgrowth of the *Candida* [24]. Lastly, *Lactobacilli* cause modulation of the local immune system [39].

When the healthy microbial balance is disturbed, *Lactobacilli* may lose their ascendancy, and other microorganisms, such as *Candida albicans*, can foster and cause overgrowth. Multiple factors can alter the vaginal microbiota and disturb the balance between tolerance and invasion of *Candida* species. Important drivers for the pathogenesis of VVC are changes in the *Lactobacilli* community, elevated estrogen levels (i.e., due to oral contraceptives, hormone replacement therapy (HRT) used in post-menopause, being in the luteal phase of the menstruation cycle or pregnancy), an elevated pH, and the presence of glucose and eicosanoids (such as prostaglandin E2 and thromboxane B2). Other determinants have an inhibitory effect on VVC such as lactate and the presence of short-chain fatty acids such as acetate, butyrate, and propionate B [4,6].

### 4.2. Host-Related Predisposing Factors

A broad spectrum of host-related predisposing factors such as genetic background, (uncontrolled) diabetes mellitus, altered immune status, use of steroids, and antibiotics therapy, as well as behavioral factors such as sexual activity, hormone replacement therapy, and use of contraceptives including intrauterine devices, have been associated to promote VVC pathology [1,2,4].

### 4.3. Idiopathic RVVC

There are no predisposing factors in 20–30% of the RVVC patients. It is suggested that the *Candida* strain, its virulence, and inter-individual differences play a key role in idiopathic RVVC pathogenesis [1,4]. Several epidemiologic and cohort studies demonstrated that genetic mutations and polymorphisms and ethnicity play a role [4,31]. Moreover, NAC species are also associated with recurrent infections in VVC patients, likely because of their natural resistance towards azole-based antifungal agents [4].

## 5. Treatment of RVVC and Its Efficacy

The standard treatment for RVVC consists of 10–14 days of induction therapy with a topical antifungal agent or oral fluconazole 150 mg, followed by fluconazole 150 mg a week for 6 months [5,42]. Fluconazole is primarily fungistatic by selectively inhibiting the fungal cytochrome P450-dependent enzyme named lanosterol 14-α-demethylase, encoded by the *ERG11* gene, which converts lanosterol to ergosterol [43,44]. Ergosterol is an essential component of fungal plasma membranes, and inhibition of its biosynthesis leads to loss of membrane structural and functional properties, including altered fluidity and permeability [44,45]. Moreover, toxic methylated sterols accumulate in the fungal cell membrane via *ERG3*, and cell growth is arrested [43,46,47].

The morbidity of RVVC is dramatically increasing, and also the costs associated with medical care rise accordingly [4]. It is difficult to achieve a long-term cure. In research published in the *New England Journal of Medicine*, the proportions of women who remained disease-free at 6, 9, and 12 months after weekly treatment with fluconazole 150 mg for 6 months were 90.8%, 73.2%, and 42.9%, respectively [3]. Despite fluconazole being effective to relieve VVC symptoms, a long-term cure is hard to achieve and maintain [2,3].

### 5.1. Resistance towards Fluconazole

Since fluconazole is fungistatic rather than fungicidal, there is an increased opportunity to develop acquired resistance in the presence of this antifungal [46]. There are several challenges in fluconazole treatment such as an increase in antifungal resistance and VVC caused by NAC species, as well as the existence of biofilms [48]. Epidemiologic studies confirm that mostly all women diagnosed with fluconazole-resistant *Candida albicans* were previously exposed to fluconazole [48]. A study determined the prevalence of vaginal colonization by *Candida* in women and evaluated the susceptibility for the antifungal treatment with fluconazole [6]. There were 612 women recruited to determine the presence of *Candida* species, of which 20.1% were colonized with yeasts. In most cases, *Candida albicans* (68.3%) was found, followed by *Candida glabrata* (16.3%) and *Candida parapsilosis* (8.9%). A total of 79% of the *Candida* isolates were susceptible to fluconazole [6]. As expected, the susceptibility of this mixed population of *Candida* species is slightly lower than that of *Candida albicans* alone. The susceptibility rates of *Candida albicans* to fluconazole reported in other studies are 89.5%, 87.5%, 86.5%, 85.3%, and 95.3%, meaning about only 5–15% of *Candida albicans* strains are resistant [49,50,51,52,53]. To illustrate, resistance levels of *Candida krusei* (66.6%), *Candida glabrata* (60%), *Candida kefyr* (45.5%), and *Candida parapsilosis* (6.9%) are typically much higher [49]. Limited options remain for maintenance therapy for RVVC when azole resistance appears [1].

### 5.2. Unnecessary and Inappropriate Use of Fluconazole

Fluconazole is a commonly used antifungal agent and is easily accessible, which increases the risk of developing resistance. For example, the over-the-counter availability of antifungal agents combined with the frequent empiric prescription of fluconazole for sporadic VVC and the frequent use of a low weekly dose of fluconazole as a maintenance regimen facilitates fluconazole-resistance by *Candida* species, subsequently leading to RVVC [48]. Long-term maintenance therapy should be based on diagnostic confirmation to avoid unnecessary and inappropriate use [48].

### 5.3. Non-Albicans Species

VVC caused by NAC species is increasingly common due to overuse and misuse of antifungal agents [6,15,54]. Furthermore, a major concern regarding the increased incidence of VVC caused by NAC species is that such infections are often more difficult to treat because they are less susceptible to azoles and are more frequently resistant [6,15,24].

### 5.4. Biofilms Complicate RVVC Treatment

The biofilm formation in the pathogenesis of VVC provides elevated virulence and resistance towards antifungal therapy such as fluconazole [27,32,33]. A 1000-fold higher resistance profile of biofilms compared to their planktonic counterparts has been described [20,55,56]. Biofilms are also less sensitive to eradicate by the host immune system. A study of clinical isolates obtained from women with at least two episodes of VVC confirmed a lower antifungal susceptibility for biofilms in comparison with the planktonic antifungal susceptibility [32].

Currently, no treatment targets *Candida* biofilm formation and eradication, making biofilms a significant clinical challenge that urgently requires novel treatment options [31,33,34,48].

## 6. Medical-Grade Honey as an Alternative Treatment Option

The high recurrence rate of complaints after fluconazole treatment may be attributed to the fact that fluconazole only interacts with the yeast, hyphae, and invasive *Candida* stages (Figure 2). In contrast, when an established biofilm is present, the ECM prevents the fluconazole from reaching the *Candida* cells, and therefore it will not have an effect on biofilms [57]. Moreover, fluconazole does not affect the vaginal mucosal response [1,3,4]. Since ancient times, honey has been used for wound treatment and care because of its antimicrobial and wound healing activities. Acquired azole resistance, the epidemiological shift from *Candida albicans* to NAC species, and the existence of biofilms demand better treatment options. Medical-grade honey (MGH) could be an accessible, effective, and affordable option [24]. To assure the safety and efficacy of honey for clinical application, strict guidelines are followed to establish MGH [58]. MGH is effective in acute and chronic wounds and provides rapid epithelization and wound contraction, has anti-inflammatory activity, stimulates debridement, decreases pain, resolves infections, decreases wound healing time, and is cost-effective [59]. The use of honey for reducing biofilm formation on indwelling plastic devices such as urinary catheters are also considered, but more research is needed [60,61].

As mentioned before, four key factors determine the progression and development of RVVC: the presence of *Candida*, the population of *Lactobacilli*, the microenvironment, and host-related factors. In contrast to fluconazole, MGH may affect all these factors via multiple mechanisms (Figure 2, Table 1).

### 6.1. The Antimicrobial Activity of MGH against Candida Species

MGH has multiple physicochemical properties that result in antimicrobial and healing activities. MGH consists of more than 200 different constituents, of which the relative majority consists of carbohydrates (about 80%), such as glucose and fructose, and water (17–18%), as well as a small number of other substances such as minerals, vitamins, organic acids, enzymes, phenolic compounds, flavonoids, and other phytochemicals [58,62].

There are a couple of factors responsible for the antifungal effects of honey. The sugar-rich composition has an osmotic activity, attracting fluid from the surrounding environment, and will also lead to dehydration of present microorganisms, which makes the microorganisms vulnerable [63]. One of the strongest antifungal factors of MGH is hydrogen peroxide [62,63,64]. When the sugars from the honey come into contact with water, under the presence of the enzyme glucose oxidase in MGH, hydrogen peroxide will be formed and released: C_6_H_12_O_6_ (glucose) + H_2_O + O_2_ + glucose oxidase → C_6_H_12_O_7_ (gluconic acid) + H_2_O_2_ (hydrogen peroxide) [63,65,66]. Hydrogen peroxide is a well-known antimicrobial molecule that will kill almost all microorganisms, including those resistant to antibiotics [62,63,66,67]. The acidic pH of MGH makes it even harder for most microorganisms to persist. Moreover, other molecules that are present in MGH also have a direct antimicrobial effect, such as phenolic compounds, flavonoids, methylglyoxal, and bee defensin-1 [62]. Since the antimicrobial activity of MGH is based on multiple mechanisms, microorganisms are not capable of developing resistance towards MGH [62,66,68,69,70,71].

MGH can play a beneficial role in all *Candida* stages, whereas fluconazole (when the *Candida* is not resistant) only interacts with the first three stages from yeast to invasion (except the formation of biofilms) by interfering with the plasma membrane synthesis, adhesion, and growth of the *Candida* [43,46,72]. The biofilm forms a physical shield for fluconazole, preventing the reaching of the *Candida* cells [73,74]. The activity of MGH against biofilms may therefore be a discriminating factor that results in a long-term cure. The antifungal activity of honey against *Candida* species has been widely investigated in vitro (Table 2). In these studies, species of patients with VVC were isolated, and minimal inhibitory concentrations (MIC) and minimal fungicidal concentrations (MFC) were determined, also against fluconazole-resistant species and biofilm models.

### 6.2. MGH Resolves Non-Albicans Candida Species

As noticed previously, NAC species create a major problem in the treatment of RVVC. The incidence of NAC species increases, and they demonstrate elevated resistance for antifungal agents [82]. Besides the efficacy of honey against *Candida albicans*, several studies have demonstrated the susceptibility of NAC species such as *Candida tropicalis*, *Candida glabrata*, *Candida parapsilosis*, *Candida kefyr*, and *Candida dubliniensis* to honey, which could fulfill the growing demand for new antifungal agents [62,75,81]. MGH does not distinguish between the different *Candida* species, their virulence factors, and their resistance profiles.

### 6.3. The Effect of MGH on Biofilms

The ECM of the biofilm is responsible for increased resistance, preventing antifungal agents from penetrating the biofilm and shielding the *Candida* cells from the host immune response. MGH can be the solution for this problem against the failed treatment of *Candida* biofilms in RVVC. MGH reduces the production of extracellular polysaccharide matrix, which leads to the prevention of biofilm formation and also promotes the eradication of mature biofilms [62,77].

### 6.4. Lactobacilli Are Not Affected by MGH

*Lactobacilli* maintain a healthy vaginal microbiome and are natural competitors of *Candida* in the vagina. When the concentration of *Lactobacilli* decreases, an overgrowth of *Candida* species can occur. It is important that MGH does not interrupt the beneficial effects of *Lactobacilli*, which is supported by an in vitro study confirming no interference [7,24]. Moreover, an in vivo study in rats showed that honey enhanced the growth of *Lactobacilli* [83]. Furthermore, MGH has some overlapping characteristics with *Lactobacilli* such as the production of hydrogen peroxide and maintaining a low pH, being able to replace or augment these effectors.

### 6.5. MGH Modulates the Vaginal Microenvironment

Besides the antimicrobial activity demonstrated above, MGH has anti-inflammatory and antioxidative properties that can further benefit the vaginal environment, especially under infected and inflammatory conditions [84,85]. A pro-inflammatory state of the vagina makes the tissue more prone to *Candida* infections, as is the case with RVVC [19]. Phytochemicals, such as flavonoids, polyphenolic compounds, and vitamin C, are antioxidants present in the MGH, which subsequently scavenge free oxygen radicals, reducing inflammation and minimizing tissue damage [62,86,87]. Therefore, we are confident that MGH not only will fight the infection but also will harness the tissue for new infections by attenuating the inflammatory state and skew it towards protection. 

### 6.6. Effect of MGH on the Immune Status

The pro-inflammatory and antioxidative activity of MGH may, in addition to changing the microenvironment, also affect the immune response. Leukocytes migrate in response towards cytokines and chemokines that are produced under the influence of the local microenvironment [88,89]. By influencing the microenvironment (pH, H_2_O_2_, inflammatory and oxidative state), MGH can modulate immunological mediators and affect the immune response [90]. For example, MGH may stimulate the recruitment of neutrophils and monocytes to the site of injury and may switch the phenotype of macrophages from pro-inflammatory towards anti-inflammatory, creating a ‘protective’ microenvironment [91,92]. Moreover, the immunomodulatory properties of MGH may prevent infections by stimulating the immune response in combination with their antimicrobial activity. Interestingly, a prophylactic activity of an MGH-based wound care formulation has previously been found in randomized controlled trials, investigating a single subcutaneous application in both equine lacerations and colic surgeries [93,94]. In these studies, the infection rate decreased three- and fourfold, respectively.

### 6.7. Clinical Studies Supporting a Role for MGH in the Treatment of (R)VVC

MGH has shown beneficial effects in the treatment of (R)VVC. Several studies confirmed a potent role for honey in the treatment of (R)VVC. Most studies demonstrated a significant decrease in inflammation, itching, and discharge. A summary of these studies is presented in Table 3.

### 6.8. Considerations for MGH Application

In contrast to fluconazole, MGH exerts pleiotropic effects and acts via multiple pathways, as mentioned above. This may be beneficial for the treatment of RVVC as it can act on multiple relevant targets. For example, the multitude of antimicrobial mechanisms ensures effective killing of the different *Candida* species, irrespective of their resistance profile. These different mechanisms lead to dehydration of the fungal cells, weakening of the cell membranes, and causing of intracellular damage, resulting in limiting its reproduction and causing cell death, while simultaneously preventing the risk of resistance. Please note that some of the proposed targets demand further investigation. For example, more research on the effect of MGH on the *Lactobacilli* population and the influence on the pH and lactate and hydrogen peroxide production in a complete ecosystem is necessary. Previous studies on honey-containing products for the treatment of vaginal infections reported it to be safe and without serious side effects (besides non-compliance (6%), soiling of underclothes (17%), and local irritation (1.2%)) [98,99]. It is not advised to use honey-based products on patients with known allergy to honey, and future research should further investigate potential side effects.

## 7. Study Design of a New Prospective Randomized Controlled Trial

Several small studies recommend performing a study with a larger sample size, longer duration of treatment, and a longer follow-up [95,96,98,99]. A prospective randomized controlled trial to compare the efficacy of fluconazole (Diflucan^®^) and an MGH formulation (L-Mesitran Soft^®^) for the treatment of recurrent vulvovaginal candidiasis will start this year. The trial is registered on ClinicalTrials.gov (NCT04626258) and the Dutch National Clinical Trial database (NL 73974.000.21). The primary objective is to investigate the mycological cure of L-Mesitran Soft in relation to the current standard of care (fluconazole), 1 month after starting treatment in patients with RVVC, defined as at least one positive *Candida* culture and a minimum of three episodes of symptoms in one year. Since the number of relapses is important for investigating the long-term efficacy, the follow-up period will be 12 months. Secondary objectives are to investigate the effects on the clinical cure and symptoms, including redness, irritation, itching, dysuria, dyspareunia, and vaginal discharge. In addition, the prophylactic activity after 6 months of maintenance therapy and the number of relapses within 12 months will be investigated for long-term efficacy. Moreover, information about the side effects, discomfort, and quality of life will be collected and compared. Vaginal swabs will be collected at 1 month, 6 months, and 12 months after starting treatment. The patients will also fill in a questionnaire at inclusion and 1, 6, 9, and 12 months after the start of the therapy. The interventions are fluconazole (oral intake of 150 mg capsules) at days 1, 4 and 7, followed by weekly intake for six months and L-Mesitran Soft (intravaginally 5 g with applicator) daily for 1 month, followed by weekly intravaginal application for the next 5 months. For fluconazole, this is the standard therapy regime [42]. For L-Mesitran Soft, a daily dosage of 5 g MGH is frequently used in literature, as is also shown in Table 3 [95,96,97,100].

## 8. Rationale for Selecting the MGH-Based Formulation

Many different types of honey exist, varying in composition and antimicrobial activity, e.g., due to floral composition and spatiotemporal differences. A limited number of MGH-based formulations that are CE- and FDA-approved exist (Medihoney, Activon, Manuka Fill, and L-Mesitran). Most of these products contain manuka honey, of which the main antimicrobial mechanism is based on methylglyoxal rather than hydrogen peroxide. We selected L-Mesitran for our study because this MGH-based formulation acts via hydrogen peroxide, and this may be more relevant for this indication. Moreover, several direct comparison studies show that other types of honey and L-Mesitran have stronger antimicrobial activity and enhanced efficacy against mucositis than manuka honey [103,104,105,106,107]. Moreover, L-Mesitran ointment has previously been used in a pilot study (*n* = 30) for the treatment of clinical vaginal infections, presenting a positive effect on the clinical and mycological cure [108].

The L-Mesitran Soft formulation consists of 40% MGH and is supplemented with other ingredients: medical-grade lanolin, PEG4000, propylene glycol, and vitamins C and E. Multiple studies have directly compared L-Mesitran Soft with its raw honey component, including against different fungi (*Malassezia pachydermatitis*, *Candida albicans*, *Candida glabrata*, *Candida krusei*, *Candida parapsilosis*, and *Candida auris*) [79,103,109,110]. Interestingly, all of these studies showed that the complete formulation had a stronger antimicrobial activity when compared to the raw honey component [79,103,109,110]. MGH also demonstrated a synergistic activity with the other ingredients of L-Mesitran Soft on the inhibition and eradication of biofilms [104]. Moreover, L-Mesitran Soft is effective in treating chronic wounds infected with (multi-resistant) bacteria, even when present in biofilms [65,104,111]. There is extensive evidence of the antimicrobial and wound healing properties of L-Mesitran Soft in literature, and no negative side effects or safety issues have been reported [65,68,69,70,71,104,111,112]. This should ease the registration of the product for new indications, such as RVVC, when the trial is successful.

## 9. Conclusions

The population of women with RVVC is estimated to increase over the coming years. The most commonly used treatment for RVVC is fluconazole, which it seems will become inadequate in the future due to increasing resistance to fluconazole, the rise of NAC species, and the existence of biofilms. MGH might be a promising treatment in VVC and RVVC. In contrast to fluconazole, MGH is expected to have multiple beneficial mechanisms. In addition to the antifungal activity on *Candida*, MGH can likely also eliminate antifungal-resistant *Candida* species, including NAC species, and eradicate biofilms. Moreover, MGH can modulate the microenvironment by its anti-inflammatory and antioxidative activity. In vitro studies and clinical studies demonstrated honey as a promising alternative therapy for RVVC. More research is needed to investigate the exact clinical efficacy and the long-term cure rate of MGH; as proposed, this will be further investigated in a randomized controlled trial.

## Figures and Tables

**Figure 1 jof-07-00664-f001:**
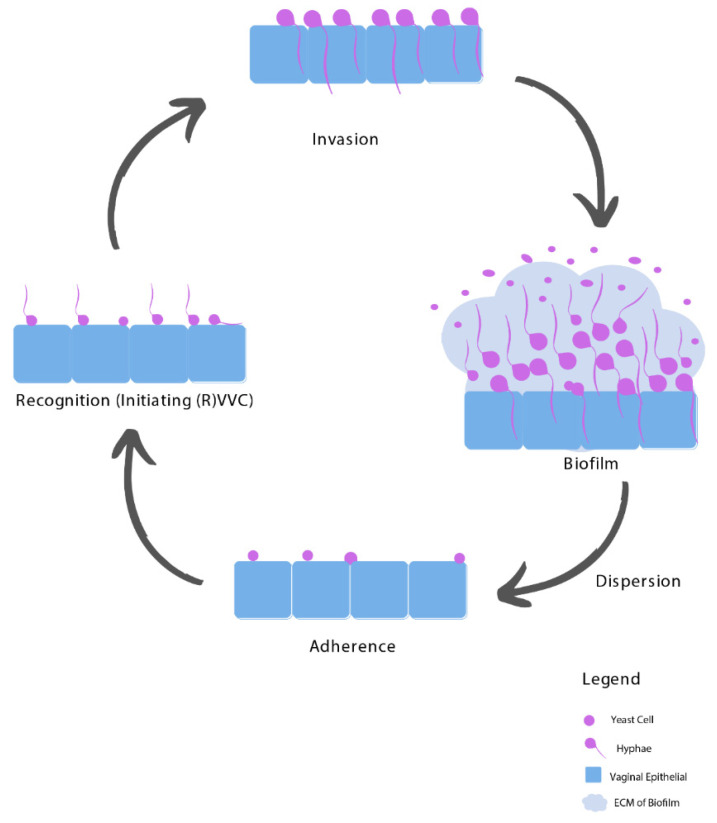
Biofilm formation of Candida species during the pathogenesis of (R)VVC. Yeast cells adhere to the vaginal epithelium and subsequently form hyphae, which together with secretion of Candidalysin determine recognition by the immune system and drive damage and vulvovaginal immunopathogenesis. Invasion of the *Candida* follows through hyphal extension, relying on both the strain and host factors. Next, biofilms can start to form, from which *Candida* cells can be released (dispersion) as planktonic cells and reinitiate the complete process.

**Figure 2 jof-07-00664-f002:**
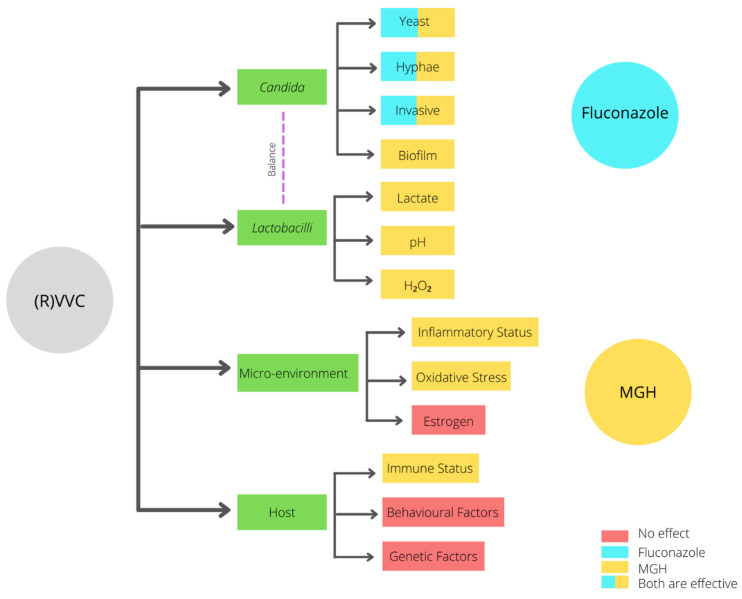
Differential antipathogenic activity of fluconazole and MGH on the pathogenesis of RVVC.

**Table 1 jof-07-00664-t001:** Overview antipathogenic activity of fluconazole and MGH.

Characteristic	Fluconazole	MGH
*Candida albicans*	+	+
(Increased raise in VVC caused by) NAC species	−	+
Biofilms	−	+
Increased resistance	−	+
Microenvironment/vaginal mucosal response	−	+
*Lactobacilli*	−	+−
pH	−	+
Osmotic effect	−	+
Antimicrobial	+	+
Anti-inflammatory	−	+
Antioxidative	−	+

−: no effect; +: positive effect; +−: possible effect.

**Table 2 jof-07-00664-t002:** Overview in vitro studies investigating the antimicrobial activity of MGH against *Candida* species.

Selected Studies	Methods	Results
Irish et al. 2006 [75]	Clinical isolates of *Candida albicans*, *Candida glabrata,* and *Candida dubliensis* were tested against four different honeys: -Jarrrah honey with hydrogen peroxide activity -Medihoney -Comvita wound care 18+, a pure Leptospermum honey -Artificial honey used to simulate the high sugar levels found in honey	-Jarrah honey has a MIC of 18.5% compared to 38.2% for Medihoney, 39.9% for Comvita honey, and 42.6% for artificial honey against *Candida albicans*. Jarrah honey was also significantly more active against *Candida Glabrata* and *Candida dubliensis*. -Honey was effective against isolates who are resistant to itraconazole and/or fluconazole.
Shokri et al. 2017 [76]	Fungicidal efficacy of 3 Iranian honey samples against fluconazole-resistant *Candida* species (including *Candida albicans, Candida glabrata, Candida krusei*, and *Candida tropicalis*) isolated from HIV+ patients with candidiasis	All tested honeys had antifungal activity against FLU-resistant *Candida* species, ranging from 20% to 56.25% (*v/v*) and 25% to 56.25% (*v/v*) for MICs and minimum fungicidal concentrations (MFCs), respectively. No statistically significant differences were observed between the honey samples.
Ansari et al. 2013 [77]	Effect of jujube honey on *Candida albicans* growth and biofilm formation	40% *w/v* Jujube honey interferes with formation of *Candida albicans* biofilms and disrupts established biofilms.
Banaean-Boroujeni et al. 2013 [7]	Effect of Iranian honey and miconazol against *Candida albicans*, in vitro	Honey prevented growth of *Candida albicans* greatly only at an 80% concentration, whereas miconazol inhibited it completely. Honey did not interfere with the *Lactobacillus.*
Estevinho et al. 2011 [78]	Monofloral lavender honey samples were analyzed to test antifungal effect against *Candida albicans, Candida krusei*, and *Cryptococcus neoformans*	The honey concentration that inhibited 10% of the yeasts’ growth ranged from 31% (*Candida albicans*) and 16.8% (*Candida krusei*).
Koc et al. 2009 [67]	Antifungal activity of four Turkish honey samples against 40 yeast species (*Candida albicans, Candida Krusei, Candida Glabrata*, and *Trichosporon* spp.)	The honeys had antifungal activity for *Candida albicans* at a mean MIC of 45.56%, for *Candida Glabrata* at 64.09%, and for *Candida krusei* at 56.88%.
Hermanns et al. 2019 [79]	Five clinical *Candida albicans* isolates and a control strain were tested against unprocessed Mexican Yucatan honey with hydrogen peroxide activity and L-Mesitran soft	-No effect of 40% Mexican Yucatan MGH alone. -L-mesitran has fungistatic (25–50% MIC) and fungicidal (50% MFC) activity, corresponding to honey concentrations of 10% and 20%, respectively. -The supplements in L-Mesitran enhanced the antimicrobial activity of the honey formulation.
Al-Waili et al. 2005 [80]	Effects of honey, olive oil, and beeswax and the mixture on growth of Staphylococcus aureus and *Candida albicans* isolates	The amount of honey present in the honey mixture (50% *w/v*) completely inhibited the growth of *Candida albicans*.
Khosravi et al. 2008 [81]	Anti-candidal activity of 28 locally produced honeys from Iran against *Candida albicans, Candida parapsilosis, Candida tropicalis, Candida kefyr, Candida glabrata*, and *Candida dubliniensis*	The MIC and MFC means of different honeys were 24–47% and 29–56% against different *Candida* species, respectively.
Fernandes et al. 2021 [62]	Antifungal activity of five Portuguese honeys and Manuka honey in planktonic and biofilm models of *Candida albicans, Candida tropicalis, Candida glabrata*, and *Candida parapsilosis*	-All honeys had a potent activity against *Candida* species (MIC 25–50% (*w/v*)). -Biofilms can be reduced at a concentration of 50–75% honey.

**Table 3 jof-07-00664-t003:** Overview clinical studies investigating the effect of honey on VVC. RCT: randomized controlled trial.

Selected Studies	Treatment	Trial Design	Purpose/Objective/Content	Results/Conclusion
Banaeian et al. 2017 [95]	Clotrimazole 1% cream (*n* = 36) versus honey cream (honey and neutral crème in 70:30 ratio) (*n* = 44), both 5 g with applicator for 7 nights	RCT	-Inflammation -Vaginal discharge -Irritation/itching At baseline in the fourth and eight days of treatment -Recurrence after 3 months	-Significant decrease in discharge, inflammation and itching in both groups. -Inflammation and discharge scores were significantly lower on the eight day in the clotrimazole group, compared to the honey group. -No significant difference in the itching score on the eighth day between the honey and clotrimazole groups. -Greater efficacy of honey for RVVC. After 3 months, 5 of 17 patients from the honey group had a recurrent infection, and 8 of 15 patients of the clotrimazole group had a recurrent infection. -Honey decreased VVC symptoms without affecting *lactobacillus.*
Seifinadergoli et al. 2018 [96]	Clotrimazole 1% cream (*n* = 53) versus 50% honey gel (*n* = 53), both 5 g with applicator for 8 nights	RCT	-Vaginal discharge -Itching -Burning -Dyspareunia -Urinary problem At baseline, in the fourth and eight days after treatment	-Vaginal discharge for 68% in honey group versus 72% in clotrimazole group after 8 days (*p* < 0.001). -Itching for 4% in honey group versus 10% in clotrimazole group after 8 days (*p* < 0.001). -Burning for 4% in honey group versus 0% in clotrimazole group after 8 days (*p* < 0.001). -Culture was 50% positive in honey group versus 20% in clotrimazole group after eight days, and there was no statistical difference. -Equally effective in making the culture and wet smear results negative. -Honey can be used as an alternative for other antifungal drugs.
Darvishi et al. 2015 [97]	Clotrimazole 1% cream (*n* = 35) versus cream mixed of yogurt and honey (*n* = 35), both 5 g with applicator for 7 days	RCT	-Itching -Irritation -Dysuria -Dyspareunia -Discharge -Cultures At baseline, 7 days after treatment, 14 days after treatment	-Itching for 2.9% in yogurt and honey group versus 25.7% in clotrimazole group 14 days after treatment (*p* < 0.02). -Irritation for 2.9% in yogurt and honey group versus 22.9% in clotrimazole group 14 days after treatment (*p* < 0.04). -Discharge for 2.9% in yogurt and honey group versus 22.9% in clotrimazole group 14 days after treatment (*p* < 0.04). -Significant improvement in symptoms in yogurt and honey group compared to clotrimazole (*p* < 0.05). -No significant differences in mycological cure rate between yoghurt and honey (71.4%) and clotrimazole (85.7%) groups at 14 days after treatment.
Abdelmonem et al. 2012 [98]	Tioconazole 100 mg vaginal tablet (*n* = 47) once daily for 7 days versus bee honey and yogurt mixture vaginally with an applicator 30 g 62.5% honey twice a day for 7 days (*n* = 82) by pregnant women	Prospective comparative study	-Clinical cure -Mycological cure Before treatment and after treatment	-Clinical cure rate was significantly higher in the honey and yogurt group (87.8%) compared to the tioconazole group (72.3%). -The mycological cure rate was significantly higher in the tiozonazole group (91.5%) compared to the honey and yogurt group (76.9%). -Itching was 3.7% in the honey and yogurt group versus 14.9% in the tioconazole group. -Discharge was 4.9% in the honey and yogurt group versus 12.8% in the tioconazole group. -Vulvo-vaginal redness was 3.7% in the honey and yogurt group versus 8.5% in the tioconazole group.
Aboushady et al. 2015 [99]	Fluconazole (*n* = 30) versus Sider honey 5 mL (80%) (*n* = 30) applied vaginally twice/day for 7 days	Quasi-experimental design	-Clinical cure -Mycological cure Before treatment and after treatment	-Clinical cure rate from sider honey (86.6%) compared to fluconazole (40%). -Mycological cure rate from sider honey (76.7%) compared to fluconazole (43.3%). -Vaginal discharge after sider honey (10%) compared to fluconazole (50%). -Burning sensation after sider honey (0%) compared to fluconazole (33.3%). -Itching after sider honey (0%) compared to fluconazole (10%).
Jahdi et al. 2021 [100]	-Mixture of honey and yogurt cream (*n* = 35) -Honey cream (*n* = 35) -Clotrimazole cream (*n* = 35) versus mixture of honey and yogurt cream (*n* = 35) versus honey cream (*n* = 35); each group was treated for 7 nights, all 5 g with applicator intravaginally	RCT	-Leucorrhea -Itch -Irritation -Dysuria -Dyspareunia Before, at 7 and 14 days after treatment -Mycological cure rate 7 and 14 days after treatment	-Itching after 14 days of treatment with clotrimazole (25.7%) versus yogurt and honey (2.9%) versus honey (14%). -Irritation after 14 days of treatment with clotrimazole (22.9%) versus yogurt and honey (2.9%) versus honey (5.7%). -Significant difference between yogurt and honey group and honey vaginal cream group versus clotrimazole group in improving symptoms. -Negative culture 14 days after treatment was found in yogurt and honey group (71.4%), honey group (68.6%), and clotrimazole group (85.7%) with no significant differences (*p* > 0.05).
Rasooli et al. 2019 [101]	-Honey cinnamon vaginal cream (*n* = 50) -Clotrimazole vaginal cream (*n* = 50)	RCT	-Clinical effect -Microscopic effect	-In both groups, symptoms improved after treatment (*p* < 0.001). -In the honey cinnamon vaginal cream group, burning was significantly less than the clotrimazole group (*p* = 0.008). -Culture results were similar after treatment (*p* = 0.461). -Honey cinnamon vaginal cream can be advisable to use it as an alternative to the clotrimazole vaginal cream.
Fazel et al. 2017 [102]	-Honey 5 mL intravaginally (*n* = 15) -Honey 5 mL and clotrimazole 100 mg (*n* = 30) -Clotrimazole 100 mg intravaginally (*n* = 32) -Treated daily for 7 days	Double blind clinical trial	-Clinical effect -Microscopic effect	-Alleviations occurred in symptoms and signs of vaginitis after treatment with honey and clotrimazole. -The success rate was 100% in the honey group and in the honey and clotrimazole group. -Use of honey alone represents a novel and effective formulation for the treatment of vaginitis.

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
