# Peer review of "Treating (Recurrent) Vulvovaginal Candidiasis with Medical-Grade Honey—Concepts and Practical Considerations"

_jof, 2021, doi:10.3390/jof7080664_

Round 1

Reviewer 1 Report

In this review, authors discuss the rationale and provide arguments in favor of using Medical Grade Honey (MGH) as an alternative therapeutic strategy for treating (recurrent) vulvovaginal candidiasis. Although interesting, the rationale for using MGH instead of azoles is not clear and needs a deeper analysis of the literature. In addition, the description of some background on C. albicans biology/molecular pathogenesis in the context of (R)VVC is lacking and/or superficial (if not saying simplistic). Extensive revision of English language is required. 

Specific comments:

-Extensive revision of English is required. 

-Figure 1 rather describes biofilm formation. Description of the mechanism in legend to figure 1 is not correct: "(...)which is the first stage to become pathogenic", hyphal development is not necessary the first step for C. albicans to "become pathogenic". "Hyphae can become invasive" is not a correct statement. Candida albicans invades tissues through hyphal extension and the process is quite complex, relying on both C. albicans and host factors. Please rephrase figure legend for more clarity and read: https://www.ncbi.nlm.nih.gov/pmc/articles/PMC5778364/ (candidalysin).

-Figure 2 gives the impression that MGH exerts pleiotropic effects, which is not a favorable argument in line with its use as a therapeutic approach for VVC (at least, as shown in figure 2). Same impression when examining Table 1.

-Paragraph #6.6 (lines 346-353) has nothing to do with immunity/immune status.

Minor points:

-Line 45-46: "In 2016, RVVC affected about 138 million women per year worldwide": this statement looks confusing. Please rephrase for more clarity (Is it "138 million cases occurring in 2016" rather than "138 million per year on average"?)

-Line 49: "increaseS" (s missing).

-Line 75: Chromogenic agar does help identifying species, but still with some uncertaintly, especially with regard to differentiation between C. albicans and C. dubliniensis (both displaying blue colonies). Molecular methods (e.g. ITS sequencing, MALDI ToF) are used to confirm species even if some ambiguity may arise from MALDI ToF data. Sabouraud Glucose is not a selective/identification medium, as it allows growth of virtually all Candida species.

-Lines 92-101: Use the term "virulence trait" rather than "virulence factor" to describe a phenotypic process such as a morphological change (e.g. filamentous growth or biofilm formation) involved in the interaction of Candida with the host environment. 

-Lines 173-186: This section of the paragraph is difficult to read.

-Line 188: "microorganisms as Candida albicans can cause Candida overgrowth": not clear. How does a microorganism cause overgrowth of itself? Please rephrase for more clarity.

-Line 215: Remove entire line 215 as it is misleading with regard to the precise mode of action of azoles.

-Lines 343-345 are copy-pasted in lines 348-350 (or authors forgot to remove one of them).

Author Response

Dear reviewer 1,

Thank you for your clear and constructive comments. We have adjusted the manuscript accordingly. We are confident that the quality of the manuscript has much improved. Below a point-by-point response is provided. In the attachment, the latest version of the manuscript can be found with "tracked changes".

Specific comments:

-Extensive revision of English is required. 

A: English language and style are checked by a native speaker and extensive grammar changes have been made.

-Figure 1 rather describes biofilm formation. Description of the mechanism in legend to figure 1 is not correct: "(...)which is the first stage to become pathogenic", hyphal development is not necessary the first step for C. albicans to "become pathogenic". "Hyphae can become invasive" is not a correct statement. Candida albicans invades tissues through hyphal extension and the process is quite complex, relying on both C. albicans and host factors. Please rephrase figure legend for more clarity and read: https://www.ncbi.nlm.nih.gov/pmc/articles/PMC5778364/ (candidalysin).

A: The Figure legend has been adapted.

-Figure 2 gives the impression that MGH exerts pleiotropic effects, which is not a favorable argument in line with its use as a therapeutic approach for VVC (at least, as shown in figure 2). Same impression when examining Table 1.

A: Thank you for your comment. In our opinion, a pleiotropic activity is beneficial for the treatment of RVVC as it can act on multiple relevant targets. However, we agree that for registrational purposes as a drug this may complicate things. Please note that we also updated Figure 2 by a designer to make it more attractive and easier to understand.

-Paragraph #6.6 (lines 346-353) has nothing to do with immunity/immune status.

A: Paragraph 6.6 is revised.

Minor points:

-Line 45-46: "In 2016, RVVC affected about 138 million women per year worldwide": this statement looks confusing. Please rephrase for more clarity (Is it "138 million cases occurring in 2016" rather than "138 million per year on average"?)

A: You are correct that it was confusing. As it is an estimation over a certain period, it is more correct to remove “In 2016”. We also inserted the original reference after this statement.

-Line 49: "increaseS" (s missing).

A: Adapted.

-Line 75: Chromogenic agar does help identifying species, but still with some uncertaintly, especially with regard to differentiation between C. albicans and C. dubliniensis (both displaying blue colonies). Molecular methods (e.g. ITS sequencing, MALDI ToF) are used to confirm species even if some ambiguity may arise from MALDI ToF data. Sabouraud Glucose is not a selective/identification medium, as it allows growth of virtually all Candida species.

A: We have added that culturing is not the most selective method and molecular methods are needed for confirmation. Citations to ITS sequencing and MALDI-ToF for fungal identification are added. To further differentiate between Candida species, additional culturing is needed, e.g., with chromogenic agar or Sabourad’s Dextrose Agar. However, culturing is not the most selective procedure, and molecular methods such as sequencing of the internal transcribed spacer (ITS Sequencing) and matrix-assisted laser desorption ionization time-of-flight mass spectrometry (MALDI-ToF MS) are needed to identify the specific species.[13,14]

-Lines 92-101: Use the term "virulence trait" rather than "virulence factor" to describe a phenotypic process such as a morphological change (e.g. filamentous growth or biofilm formation) involved in the interaction of Candida with the host environment. 

A: Adapted.

-Lines 173-186: This section of the paragraph is difficult to read.

A: This section has been adapted.

-Line 188: "microorganisms as Candida albicans can cause Candida overgrowth": not clear. How does a microorganism cause overgrowth of itself? Please rephrase for more clarity.

A: Adapted. When the healthy microbial balance is disturbed, Lactobacilli may lose their ascendancy and other microorganisms, such as Candida albicans, can flourish and cause overgrowth.”

-Line 215: Remove entire line 215 as it is misleading with regard to the precise mode of action of azoles.

A: We removed the line “This process is necessary for fungal cell wall synthesis.” and revised the rest of the text about the mode of action upon instruction of the second reviewer. “Fluconazole is primarily fungistatic by selectively inhibiting the fungal cytochrome P450-dependent enzyme named lanosterol 14-α-demethylase, encoded by the ERG11 gene, which converts lanosterol to ergosterol.[38,39] Ergosterol is an essential component of fungal plasma membranes, and inhibition of its biosynthesis leads to loss of membrane structural and functional properties, including altered fluidity and permeability.[39,40] Moreover, toxic methylated sterols accumulate in the fungal cell membrane via ERG3, and cell growth is arrested.[38,41,42]

-Lines 343-345 are copy-pasted in lines 348-350 (or authors forgot to remove one of them).

A: Correct, we forgot to remove lines 348-350. Thank you.   

Reviewer 2 Report

This review provides an overview of the burden of Recurrent Vulvovaginal Candidiasis (RVVC) and the biological traits that support Candida species pathogenicity. The paper summarizes clinical management and diagnosis of vaginal candidiasis, typical Candida virulence traits, microbiome context, and current treatment impairments. Medical Grade Honey (MGH) is then introduced as a possible alternative treatment based on current knowledge of its antimicrobial properties and safety profile as determined by previous studies. 
This structure is well organized and makes a decent overview of the properties of MGH that can make it a possible antimicrobial agent. There are numerous instances where the information must be conveyed more accurately or thoroughly. In some sections, it would be helpful to provide a broader scientific exposition of the facts, as some situations are at risk of expressing incomplete or erroneous messages. Likewise, some references should be chosen more carefully.         

Major comments:

-Lines 18-19 - "Clinical trials have demonstrated that honey can benefit bacterial and Candida-mediated vaginal infections". This sentence appears to indicate that honey contributes to the development of microbial infections and should be rephrased. 

-Lines 57-58 - The increase in resistant isolates is felt in other Candida species more significantly than C. albicans. Actually, the increasing resistance among NAC is more pressing, including the emergence of multi-drug resistance which is residual in C.  albicans. 

-Lines 215-216 - The ergosterol biosynthesis pathway is required for plasma membrane homeostasis, not cell wall. Reference 36 is not acceptable. There is extensive literature on how azole antifungals lead to ergosterol depletion and accumulation of toxic sterol intermediates in the membrane as their antifungal mode of action.

-Lines 230 - 235 - I am not convinced that the information is conveyed properly. First, 79% of the isolates being susceptible to fluconazole does not seem comparable with the other susceptibility rates reported here for C. albicans, which are up to 10% more. Second, the rate of 79% of susceptibility refers to a group of isolates from multiple Candida species, therefore the data cannot be directly compared with C. albicans susceptibility alone. This being said, the lower rate of susceptibility is likely due to the presence of NAC, which probably increases the resistance levels vs looking at C. albicans alone.

-Lines 303-304 - Fluconazole targets plasma membrane biosynthesis, not cell wall. Please refer to the abundant scientific literature on this topic. There a few studies probing a correlation between fluconazole and cell wall, but a direct effect of fluconazole on the cell wall structure is far from corroborated. Azoles target Erg11, leading to ergosterol depletion. Lack of ergosterol, being the major component of the fungal plasma membrane, leads to loss of membrane structural and functional properties, including altered fluidity and permeability; further exacerbated by the introduction of toxic sterol intermediates in the membrane via Erg3. Since most cell wall proteins are actually GPI-anchored in the membrane, it is reasonable to assume that a defective membrane can have a negative impact on the cell wall. However, this represents an indirect effect of fluconazole at best. Referring that ergosterol is required for cell wall synthesis (Lines 215-216) and that fluconazole interferes with cell wall synthesis (lines 303-304) without mentioning its actual established and direct mode of action (plasma membrane disruption) is misleading. Furthermore, it seems puzzling to debate antifungal impact in cell wall synthesis without mentioning echinocandins, which are the actual drugs that directly target the cell wall by impairing beta-glucan biosynthesis. 

-Lines 379 - 381 - If possible, please disclose the rationale behind the dosage chosen for MGH. This would be important to compare the future study results against fluconazole, especially since the authors claim, also based on previous studies, that it may represent a better treatment option than fluconazole therapy. How was the decision made to apply MGH daily for the first month and weekly after that, considering fluconazole treatment will be 1 application per month? This represents a significantly distinct treatment regimen. Also for future note, how are the authors planning to draw conclusions on the possible effects of MGH? Is the better infection clearance outcome (if it exists) due to the effect of MGH, or simply due to the increased periodicity/greater dosage of an antimicrobial compound (ie. indirect effect)? For example, in the first month alone, the results to be compared would be a treatment applied 1x vs a treatment applied 30x in the same time period. 

-Antimicrobial activity against biofilms appears to be one of the greatest advantages of MGH. Biofilms/pathogen adhesion in indwelling medical devices can be a relevant source of nosocomial infections. The possibility of coating materials with MGH components, or the treatment of medical devices with MGH preparations, could be discussed or put forth for further studies in the field. 

-The authors should introduce the MGH in more detail. The use of MGH as a possible alternative antifungal treatment represents a repurposing of an already existing formulation. What is the current application of MGH in clinical practice? Also, since this is an already existing product, the authors could strengthen the case for its new application by mentioning it can be fast-tracked since its safety profile for other applications has been verified already. 

General comments:

-Line 88 - VVC can cause microbial imbalance, but it should be mentioned that it can rather be a consequence of an underlying cause of imbalance (e.g. antibiotic therapy) that allows Candida species to overgrow.

-Lines 92-94 - What about Candida species that do not form hyphae? They can also act as opportunistic pathogens.

-Lines 95-96 - As stated in lines 97-101, virulence factors also contribute to tissue invasion accompanied by damage to epithelial cells (e.g. SAPs, candidalysin). The formation of hyphae is required (in the species that show dimorphic growth) for tissue invasion and also contributes to immune escape from phagocytes, but stating that protection from immune clearance subsequently creates tissue damage seems like an oversimplification and mixing of concepts.

-Lines 121-122 - The last sentence in the section needs rephrasing.

-Lines 126-127 - What is the idea to be conveyed here? Species emerge through an evolutionary process, not recognition pathways. Needs rephrasing for clarification.

-Lines 159-161 - Briefly indicate the composition of the ECM.

-Lines 170-172 - Replace "denominated by Lactobacilli" for "mainly Lactobacilli", as these are the major but not the only bacteria in the vaginal tract.

-Lines 401-402 - MGH showed synergistic activity against biofilms with what other compound(s)? 

-Please standardize the spelling of words such as "anti-fungal" to "antifungal", "anti-microbial" to "antimicrobial", "anti-oxidant" to "antioxidant", "Candida strains" to "Candida species" (strains refer to distinct variants of the same species), species names should be spelled in minor case letters. 

-The paper claims that MGH has antioxidant activity in the vaginal microenvironment. At the same time, it claims that MGH potentiates H2O2 production by Lactobacilli in the vaginal tract. How are these two effects combined and what is the end result?

Minor comments:

-Line 102 - Consider rephrasing to "goes through several stages".

-Lines 140-141 - Change "endocytosis and active penetration" for "endocytosis or active penetration".

-Line 315 - Replace "like" for "such as".

-Lines 343-345 are duplicated in lines 348-350.

-Table 3 - Specify the meaning of RCT in the table header.

Author Response

Thank you for your clear and constructive comments. We have adjusted the manuscript accordingly. We are confident that the quality of the manuscript has much improved. Below a point-by-point response is provided. In the attachment, the latest version of the manuscript can be found with "tracked changes".

Major comments:

-Lines 18-19 - "Clinical trials have demonstrated that honey can benefit bacterial and Candida-mediated vaginal infections". This sentence appears to indicate that honey contributes to the development of microbial infections and should be rephrased. 

A: Thank you, we have added “the treatment of” to the sentence. Clinical trials have demonstrated that honey can benefit the treatment of bacterial and Candida-mediated vaginal infections.”

-Lines 57-58 - The increase in resistant isolates is felt in other Candida species more significantly than C. albicans. Actually, the increasing resistance among NAC is more pressing, including the emergence of multi-drug resistance which is residual in C.  albicans. 

A: You are correct. We have changed “albicans” into “species”, and added that multidrug resistance is emerging. “Notably, there is increasing resistance of Candida species towards antifungal agents, making multidrug resistance emerging, while the long-term efficacy of antifungal agents is limited.[3,6,7]

-Lines 215-216 - The ergosterol biosynthesis pathway is required for plasma membrane homeostasis, not cell wall. Reference 36 is not acceptable. There is extensive literature on how azole antifungals lead to ergosterol depletion and accumulation of toxic sterol intermediates in the membrane as their antifungal mode of action.

A: We have removed “This process is necessary for fungal cell wall synthesis.” and further adapted the text. Fluconazole is primarily fungistatic by selectively inhibiting the fungal cytochrome P450-dependent enzyme named lanosterol 14-α-demethylase, encoded by the ERG11 gene, which converts lanosterol to ergosterol.[38,39] Ergosterol is an essential component of fungal plasma membranes, and inhibition of its biosynthesis leads to loss of membrane structural and functional properties, including altered fluidity and permeability.[39,40] Moreover, toxic methylated sterols accumulate in the fungal cell membrane via ERG3, and cell growth is arrested.[38,41,42]

-Lines 230 - 235 - I am not convinced that the information is conveyed properly. First, 79% of the isolates being susceptible to fluconazole does not seem comparable with the other susceptibility rates reported here for C. albicans, which are up to 10% more. Second, the rate of 79% of susceptibility refers to a group of isolates from multiple Candida species, therefore the data cannot be directly compared with C. albicans susceptibility alone. This being said, the lower rate of susceptibility is likely due to the presence of NAC, which probably increases the resistance levels vs looking at C. albicans alone.

A: Thank you for your critical observation. You are correct and we have changed the text accordingly. “There were 612 women recruited to determine the presence of Candida species, of which 20.1% were colonized with yeasts. In most cases, Candida albicans (68.3%), followed by Candida glabrata (16.3%), and Candida parapsilosis (8.9%). 79% of the Candida isolates were susceptible to fluconazole.[6] As expected, the susceptibility of this mixed population of Candida species is slightly lower than that of Candida albicans alone. The susceptibility rates of Candida albicans to fluconazole reported in other studies are 89.5%, 87.5%, 86.5%, 85.3%, and 95.3%, meaning about only 5-15% of Candida albicans strains are resistant.[42-46] To illustrate, resistance levels of Candida krusei (66.6%), Candida glabrata (60%), Candida kefyr (45.5%), and Candida parapsilosis (6.9%) are typically much higher.[42] Limited options remain for maintenance therapy for RVVC when azole resistance appears.[1]

-Lines 303-304 - Fluconazole targets plasma membrane biosynthesis, not cell wall. Please refer to the abundant scientific literature on this topic. There a few studies probing a correlation between fluconazole and cell wall, but a direct effect of fluconazole on the cell wall structure is far from corroborated. Azoles target Erg11, leading to ergosterol depletion. Lack of ergosterol, being the major component of the fungal plasma membrane, leads to loss of membrane structural and functional properties, including altered fluidity and permeability; further exacerbated by the introduction of toxic sterol intermediates in the membrane via Erg3. Since most cell wall proteins are actually GPI-anchored in the membrane, it is reasonable to assume that a defective membrane can have a negative impact on the cell wall. However, this represents an indirect effect of fluconazole at best. Referring that ergosterol is required for cell wall synthesis (Lines 215-216) and that fluconazole interferes with cell wall synthesis (lines 303-304) without mentioning its actual established and direct mode of action (plasma membrane disruption) is misleading. Furthermore, it seems puzzling to debate antifungal impact in cell wall synthesis without mentioning echinocandins, which are the actual drugs that directly target the cell wall by impairing beta-glucan biosynthesis. 

A: We have changed “cell wall” into “plasma membrane”. Thank you for the detailed description of the molecular mechanism of the antifungal activity. We have expanded the background on the molecular mechanism of fluconazole (near original lines 215-216). “Fluconazole is primarily fungistatic by selectively inhibiting the fungal cytochrome P450-dependent enzyme named lanosterol 14-α-demethylase, encoded by the ERG11 gene, which converts lanosterol to ergosterol.[38,39] Ergosterol is an essential component of fungal plasma membranes, and inhibition of its biosynthesis leads to loss of membrane structural and functional properties, including altered fluidity and permeability.[39,40] Moreover, toxic methylated sterols accumulate in the fungal cell membrane via ERG3, and cell growth is arrested.[38,41,42]

-Lines 379 - 381 - If possible, please disclose the rationale behind the dosage chosen for MGH. This would be important to compare the future study results against fluconazole, especially since the authors claim, also based on previous studies, that it may represent a better treatment option than fluconazole therapy. How was the decision made to apply MGH daily for the first month and weekly after that, considering fluconazole treatment will be 1 application per month? This represents a significantly distinct treatment regimen. Also for future note, how are the authors planning to draw conclusions on the possible effects of MGH? Is the better infection clearance outcome (if it exists) due to the effect of MGH, or simply due to the increased periodicity/greater dosage of an antimicrobial compound (ie. indirect effect)? For example, in the first month alone, the results to be compared would be a treatment applied 1x vs a treatment applied 30x in the same time period. 

A: The dosage for MGH is based on literature (summarized in Table 3). We have adapted this in the text to make this rationale more clear. “The interventions are fluconazole (oral intake of 150 mg capsules) at day 1, 4 and 7, followed by weekly intake for six months and MGH (intravaginally 5 grams with applicator) daily for 1 month, followed by weekly intravaginal application for the next 5 months. For fluconazole, this is the standard therapy regime.[38] For MGH, a daily dosage of 5 grams is frequently used in literature as is also shown in Table 3.[84-86,89]

Please note that we changed Fluconazole dosage to the standard dosage for treatment and prophylaxis of recurrent vaginal candidiasis (150 mg every third day for a total of 3 doses (day 1, 4, and 7) followed by 150 mg once weekly maintenance dose for a period of 6 months. reference: www.medicines.org.uk/emc/product/6086/smpc#gref). In our hospital it used to be once monthly, but to make the study more relevant and consistent with standard therapy we decided to update our treatment protocol and follow the standards.

Regarding your question on different treatment regimen, you are correct that the first month there will be a difference in the treatment frequency, but in our expectations, patients would not mind following a daily treatment as long as it is effective and the treatment frequency is inferior to the clinical outcome. However, we will ask the patients about their opinion on the treatment in the questionnaire. Moreover, for maintenance therapy, treatment with both fluconazole and MGH will be weekly and thus the treatment frequency would be the same.  

-Antimicrobial activity against biofilms appears to be one of the greatest advantages of MGH. Biofilms/pathogen adhesion in indwelling medical devices can be a relevant source of nosocomial infections. The possibility of coating materials with MGH components, or the treatment of medical devices with MGH preparations, could be discussed or put forth for further studies in the field. 

A: We have added a sentence to introduce the potential antibiofilm activity of MGH on medical devices. “The use of honey for reducing biofilm formation on indwelling plastic devices such as urinary catheters are also considered, but more research is needed.[56,57]

-The authors should introduce the MGH in more detail. The use of MGH as a possible alternative antifungal treatment represents a repurposing of an already existing formulation. What is the current application of MGH in clinical practice? Also, since this is an already existing product, the authors could strengthen the case for its new application by mentioning it can be fast-tracked since its safety profile for other applications has been verified already. 

A: We have added more details about MGH and the existing product.

Paragraph 6: “Since ancient times, honey has been used for wound treatment and care because of its antimicrobial and wound healing activities. Acquired azole resistance, the epidemiological shift from Candida albicans to NAC species, and the existence of biofilms demand better treatment options. Medical grade honey (MGH) could be an accessible, effective, and affordable option.[21] To assure the safety and efficacy of honey for clinical application, strict guidelines are followed to establish MGH.[54] MGH is effective in acute and chronic wounds and provides rapid epithelization and wound contraction, has anti-inflammatory activity, stimulates debridement, decreases pain, resolves infections, decreases wound healing time, and is cost-effective.[55]

Paragraph 8: “Moreover, L-Mesitran Soft is effective in treating chronic wounds infected with (multi-resistant) bacteria, even when present in biofilms.[59,94] There is extensive evidence of the antimicrobial and wound healing properties of L-Mesitran Soft in literature and no negative side effects or safety issues have been reported.[59,62-65,94,101] This should ease the registration of the product for new indications, such as RVVC, when the trial is successful.”

General comments:

-Line 88 - VVC can cause microbial imbalance, but it should be mentioned that it can rather be a consequence of an underlying cause of imbalance (e.g. antibiotic therapy) that allows Candida species to overgrow.

A: We have added a sentence. Often, there is an underlying cause of the imbalance (e.g., antibiotic therapy) that allows Candida species to overgrow.[1,2,4]

-Lines 92-94 - What about Candida species that do not form hyphae? They can also act as opportunistic pathogens.

A: The text has been adapted.

“Candida species have a dual lifestyle as both a vaginal commensal and an opportunistic pathogen.[18] The switch between yeast and hyphal growth is critical for virulence, affecting numerous properties including phenotypic and biochemical properties.[19]

-Lines 95-96 - As stated in lines 97-101, virulence factors also contribute to tissue invasion accompanied by damage to epithelial cells (e.g. SAPs, candidalysin). The formation of hyphae is required (in the species that show dimorphic growth) for tissue invasion and also contributes to immune escape from phagocytes, but stating that protection from immune clearance subsequently creates tissue damage seems like an oversimplification and mixing of concepts.

A: The text has been adapted. Virulence traits are created by microorganisms and may subsequently lead to tissue damage and makes them more pathogenic.[20] Virulence traits include the ability to make a morphological switch from yeast to hyphae, modulating the expression of adhesins to help the Candida adhere to epithelial cells, the expression of invasins, the formation of biofilms, the secretion of hydrolytic enzymes and the cytolytic peptide toxin candidalysin, and the ability to escape from phagocytosis by neutrophils and macrophages.[21-24]

-Lines 121-122 - The last sentence in the section needs rephrasing.

A: The text has been adapted. In most cases, yeast cells adhere directly to the epithelial cells; however, there may also be direct adhesion of the hyphal form of Candida to epithelial cells, as hyphae grow from one epithelial cell to an adjacent epithelial cell.[25] After adherence to the vaginal epithelium, the Candida cells can form hyphae and get recognized by the immune system.[25]

-Lines 126-127 - What is the idea to be conveyed here? Species emerge through an evolutionary process, not recognition pathways. Needs rephrasing for clarification.

A: The text has been adapted. “Candida species can be recognized via two different pathways.”

-Lines 159-161 - Briefly indicate the composition of the ECM.

A: The text has been adapted. After the invasion stage, an extracellular matrix (ECM) can be developed which results in a “biofilm” that encapsulates the Candida cells. This ECM consists of exopolymeric macromolecules, including polysaccharides, proteins, lipids, and nucleic acids, which are secreted by sessile cells within the biofilm. A mature biofilm is characterized by a structured mixture of yeast-form and hyphal cells surrounded by ECM, and it provides protection against environmental challenges.”

-Lines 170-172 - Replace "denominated by Lactobacilli" for "mainly Lactobacilli", as these are the major but not the only bacteria in the vaginal tract.

A: Adapted.

-Lines 401-402 - MGH showed synergistic activity against biofilms with what other compound(s)? 

A: Adapted. MGH also demonstrated a synergistic activity with the other ingredients of L-Mesitran Soft on the inhibition and eradication of biofilms.[99]

-Please standardize the spelling of words such as "anti-fungal" to "antifungal", "anti-microbial" to "antimicrobial", "anti-oxidant" to "antioxidant", "Candida strains" to "Candida species" (strains refer to distinct variants of the same species), species names should be spelled in minor case letters. 

A: Adapted.

-The paper claims that MGH has antioxidant activity in the vaginal microenvironment. At the same time, it claims that MGH potentiates H2O2 production by Lactobacilli in the vaginal tract. How are these two effects combined and what is the end result?

A: Good question. This is quite complex and uncertain, but our thought is that overall MGH has an antioxidant activity. This is also the case in wound healing as supported by most literature.

Probably, the level of H2O2 is the discriminating factor. In wound healing, H2O2 or MGH is also not causing damage to normal skin cells (fibroblasts, keratinocytes), but does kill microorganisms. However, the factor time may also play a role.

It is actually the glucose oxidase in the honey (an enzyme that is added by the bees) that results in the production and release of low levels of H2O2 (by breaking down glucose in the sugar) and not the stimulation of H2O2 production by the Lactobacilli. Apparently, Lactobacilli are more protected against H2O2 themselves in comparison to other microorganisms as MGH has shown broad-spectrum antimicrobial activity. Likely, because they produce H2O2 themselves and therefore are used being exposed to H2O2 which makes them less sensitive. Also, in contrast to other microorganisms, Lactobacilli grow well in a low pH, making them also more resistant to a second antimicrobial mechanism of MGH. To support this, we cited several references in which MGH was not interfering with Lactobacilli.

Minor comments:

-Line 102 - Consider rephrasing to "goes through several stages".

A: Adapted.

-Lines 140-141 - Change "endocytosis and active penetration" for "endocytosis or active penetration".

A: Adapted.

-Line 315 - Replace "like" for "such as".

A: Adapted.

-Lines 343-345 are duplicated in lines 348-350.

A: Adapted.

-Table 3 - Specify the meaning of RCT in the table header.

A: Adapted.

Round 2

Reviewer 1 Report

-The authors replied on my comment on the pleiotropic effect of MGH by saying "In our opinion, a pleiotropic activity is beneficial for the treatment of RVVC as it can act on multiple relevant targets". Authors should still add a short section discussing/describing the limitations of using MGH as a potential treatment based on the fact that it could target multiple pathways (pleiotropic effect) in Candida.  

-The molecular mechanisms of VVC from the Candida side (effectors in C. albicans. e.g. Candidalysin, others) should be further developed.

Author Response

Dear reviewer 1,

Thank you for your remaining comments. We have adjusted the manuscript accordingly. Below a point-by-point response is provided. The latest version of the manuscript with tracked changes is attached.

Reviewer 1

Date of this review

26 Jul 2021 18:32:23

-The authors replied on my comment on the pleiotropic effect of MGH by saying "In our opinion, a pleiotropic activity is beneficial for the treatment of RVVC as it can act on multiple relevant targets". Authors should still add a short section discussing/describing the limitations of using MGH as a potential treatment based on the fact that it could target multiple pathways (pleiotropic effect) in Candida.  

A: We have inserted an extra paragraph to describe this more extensively.

“6.8. Considerations for MGH application                                                                                                          

In contrast to fluconazole, MGH exerts pleiotropic effects and acts via multiple pathways as mentioned above. This may be beneficial for the treatment of RVVC as it can act on multiple relevant targets. For example, the multitude of antimicrobial mechanisms ensures effective killing of the different Candida species irrespective of their resistance profile. These different mechanisms lead to dehydration of the fungal cells, weakening of the cell membranes and causing intracellular damage, resulting in limiting its reproduction and cell death, while simultaneously preventing the risk of resistance. Please note that some of the proposed targets demand further investigation. For example, more research on the effect of MGH on the Lactobacilli population and the influence on the pH, lactate and hydrogen peroxide production in a complete ecosystem is necessary. Previous studies on honey-containing products for the treatment of vaginal infections reported it to be safe and without serious side effects (besides non-compliance (6%), soiling of underclothes (17%), and local irritation (1.2%)).[98,99] It is not advised to use honey-based products on patients with known allergy to honey, and future research should further investigate potential side effects.“

-The molecular mechanisms of VVC from the Candida side (effectors in C. albicans. e.g. Candidalysin, others) should be further developed.

A: More details about the molecular mechanisms are added to paragraph 3 “Pathogenesis of RVVC”.

“This response can be triggered by host pattern recognition receptors (PRP) interacting with fungal pathogen-associated molecular patterns (PAMPs) and other more complex mechanisms (i.e., secreted aspartyl proteases (Sap)-mediated NLRP3 activation and the cytolytic peptide toxin candidalysin).[18,19] The activation of the innate immune system by a series of proinflammatory cytokines and chemokines leads to the recruitment of neutrophils to the vaginal mucosa, subsequently exacerbating symptomatic disease.[18-20] Moreover, Candida species also can exert tissue damage by direct invasion of its hyphal filaments or secretion of virulence effectors.[20]”

Reviewer 2 Report

The authors have addressed each comment and adapted the manuscript accordingly. 

I have no further observations.

Author Response

Thank you once more for your clear and constructive feedback.

We are satisfied to have addressed your questions/ comments.